# Industry 5 and the Human in Human-Centric Manufacturing

**DOI:** 10.3390/s23146416

**Published:** 2023-07-14

**Authors:** Kendra Briken, Jed Moore, Dora Scholarios, Emily Rose, Andrew Sherlock

**Affiliations:** 1Department of Work, Employment and Organisation, Strathclyde Business School, University of Strathclyde, Glasgow G4 0QU, UK; jed.h.moore@strath.ac.uk (J.M.); d.scholarios@strath.ac.uk (D.S.); 2Law School, University of Strathclyde, Glasgow G4 0LT, UK; emily.rose@strath.ac.uk; 3National Manufacturing Institute Scotland, Renfrew PA3 2EF, UK; a.sherlock@strath.ac.uk

**Keywords:** Industry 5, human–robot collaboration, interdisciplinarity, human-centric manufacturing systems, warehousing

## Abstract

Industry 4 (I4) was a revolutionary new stage for technological progress in manufacturing which promised a new level of interconnectedness between a diverse range of technologies. Sensors, as a point technology, play an important role in these developments, facilitating human–machine interaction and enabling data collection for system-level technologies. Concerns for human labour working in I4 environments (e.g., health and safety, data generation and extraction) are acknowledged by Industry 5 (I5), an update of I4 which promises greater attention to human–machine relations through a values-driven approach to collaboration and co-design. This article explores how engineering experts integrate values promoted by policy-makers into both their thinking about the human in their work and in their writing. This paper demonstrates a novel interdisciplinary approach in which an awareness of different disciplinary epistemic values associated with humans and work guides a systematic literature review and interpretive coding of practice-focussed engineering papers. Findings demonstrate evidence of an I5 human-centric approach: a high value for employees as “end-users” of innovative systems in manufacturing; and an increase in output addressing human activity in modelling and the technologies available to address this concern. However, epistemic publishing practices show that efforts to increase the effectiveness of manufacturing systems often neglect worker voice.

## 1. Introduction

Industry 4 (I4) was seen as a revolutionary new stage for technological progress in manufacturing which promised a new level of interconnectedness across a diverse range of technologies [1,2,3,4,5,6,7]. I4 refers to the digitalisation of manufacturing and involves multiple stakeholders across the lifecycle of a good or service. Sensors, as a point technology, play an important role in these developments, facilitating human–machine interaction and enabling data collection for system-level technologies. I4 focusses on progressing the machine as a learning resource [4]. In this often technology-driven innovation context, workers are regarded as human factors prone to failure, vulnerable to health and safety issues, while their skills are seen as adding value in some systems. There is growing concern, however, about the impacts of interconnected devices, data generation and extraction, and human–machine collaboration, on human labour, including employees working in newly technologically enhanced workplaces. Empirical findings show that I4 solutions tend to focus on technological progress and efficiency gains, with little to no upskilling for workers [8,9], and inconsistent, often negative, wellbeing outcomes for workers who implement such technology. Despite the best intentions, as Kinzel [10] shows for the German context, industry stakeholders admitted they had been too obsessed with technology and processes and had simply forgotten about the human factor [11]. Trade Unions supported the push towards “better, instead of cheaper” production models [12]; however, workers were kept “in the loop” only insofar as the prefigured tech-deterministic mindset driving the changes, integrated by data but rarely with their voice. Therefore, even in the German context which reflects the most sophisticated national delivery for I4 through the Platform Industrie 4.0 initiative, workers did not see the promised upskilling or increase in wellbeing. This was despite the creation of learning platforms and the influence of works councils at the company level.

This paper addresses the issues arising from these observations in engaging with I5 and the human in human-centric manufacturing and proceeds as follows: In the remainder of this Section 1, the scene is set by evaluating the newly suggested I5 conceptualisation. An interdisciplinary approach is suggested (Section 1.1) and the aligned research aims and objectives are presented (Section 1.2). Section 2 outlines the stages of the novel interdisciplinary approach in which an awareness of different disciplinary epistemic values associated with humans and work guides a systematic literature review and interpretive coding of practice-focussed engineering papers. In Section 3, the findings of the amended systematic literature review are showcased. The papers give an insight in how the human factor is represented. The focussed interpretive coding outcomes of papers scoring the highest in engaging with the human factor are outlined. The paper ends with a discussion and concluding remarks and take away points for future interdisciplinary collaborations.

The Industry 5 (I5) update promises a values-driven approach, collaboration and co-design between human and machines and attempts symbiosis to increase the effectiveness of the work system [13,14,15,16,17]. I5 reflects experiences and continuities from I4 [18,19]. An initiative driven by the European Union [14], I5 focusses on “supporting and fostering socially and ecologically relevant values” [13]: p. 5 to be integrated into industrial policy by all stakeholders involved. I5 is built upon three pillars: human centricity, resilience and sustainability [16]. The European initiative aims to strengthen specifically innovation and research in industry to remain competitive, while also tackling societal challenges for the next generation such as the green and digital transformations. In this respect, I5 policies go beyond the workplace or company level to the level of national industries and their ecosystems. Human labour here is framed around the idea that employees should be seen as an investment, not a cost. Human–machine relations in I5 are based on synergies, collaboration, empathy, trust and respect, in a “quest for value twins” as stated in the early debates on I5 by the EU Deputy Director General Ringman in 2018.


*“Although manufacturing companies are currently situated at a transition point in what has been called Industry 4.0, a new revolutionary wave—Industry 5.0—is emerging as an “Age of Augmentation” when the human and machine reconcile and work in perfect symbiosis with one another.”*
[17]

In the UK, policy stakeholders, such as the UK High Value Manufacturing Catapult outlined their vision in “Manufacturing the future workforce” [20] based on the core I5 values, with a focus on upskilling the workforce. A key pillar in the vision involves the idea of learning factories and vocational training focussed on industry needs and company-based training. The underlying assumption of these policy initiatives is that smart manufacturing leads to a replacement of tasks, or even skills, with a progression towards upskilling for employees. These visions are a work-in-progress insofar as many initiatives remain strategic, more than being backed by industry practice [21].

I5 aims to build on this push towards technological augmentation of work systems and processes by including explicit focal points around human beings and sustainability, as well as the conventional productive aspects.


*“Human-centric manufacturing is a prerequisite for future factories seeking to increase flexibility, agility and competitiveness in the face of new social challenges. The basic principle of human-centricity is that “humans should never be subservient to machines and automation, but machines and automation should be subservient to humans”*
[16,22]

As stated by the European Union and technology leaders, “(o)ne possible approach is to support interdisciplinarity of research from early on, e.g., the inclusion of social sciences in technological research. High complexity might otherwise have negative impacts on security, safety or acceptance and might slow down implementation, but fast actions are required.” [13]. Engineering, technology, life sciences and environmental sciences have collaborated organically for some time, but the inclusion of social sciences and humanities into debates or research is yet to be established.

### 1.1. Disciplinary Conceptualisations of the Human

Recognising the human factor is nothing new for engineering [15,16,23,24]. However, the human is conceptualised differently in the disciplines contributing to I5, and in their epistemic practices. In social sciences, the human at work, and by extension, human–machine relations, are usually assessed in terms of human agency using dimensions such as autonomy, discretion, skills utilisation, and employee engagement and involvement [25,26]. Work design [27] and human resource management principles [28] focus on job demand, job satisfaction and job quality [29]. Although these traditions present sometimes radically different theoretical assumptions, in principle, the human within the limits of the workplace is active, and is represented in epistemic practices in terms of worker voice. Concepts such as “High-Performance Work Systems”, for example, are understood to deliver efficiency and productivity gains through workers’ voice, which contributes to continuous improvement in the execution of the work system. Conversely, the human in natural science disciplines tends to be modelled and designed into the work system.

Industry has seen the emergence of tools and technologies which share the same physical space as workers, not only in a synchronised manner as established with assembly lines, but with the aim to “work together” with human beings. This transition challenges established frames of reference of collaboration and cooperation. For decades, the narrative for automation had seen upskilling, at least for some workforces, usually established around the notion of the operator, or symbolic analyst: the human would control and oversee the functioning of the machine based on data and experience. I4 now creates an environment where robots operate alongside human workers, connected by sensors constantly collecting data about the worker. Whether focussed on physical proximity—the robot arm stops if the human worker mistakenly comes too close—or assessment of the operator’s cognitive load, the human–machine relation has flipped. The human is in a double loop, both on the physical premises of the work environment, and in the data-loop observing and surveilling every move in real time.

### 1.2. Research Aims

References to I5 and manufacturing are increasing in the literature (SCOPUS accessed 22 March 2023, see Figure A1, Appendix A). Using the search terms Industry 5.0 AND Manufacturing with a focus on disciplines relevant for this paper, and for the timeline starting in 2017 when I5 began to be mentioned, shows multiple disciplinary contributions, with Engineering and Computer Sciences dominating (see Figure A2, Appendix A). This brief analysis of academic literature shows that, in essence, the debates promoted by industry stakeholders and policy makers, such as the EU, are now reflected in a new mindset of I5 in design and research. Historically, manufacturing always acknowledged that human labour is part of the production process, and that there often exists a tension between planning mindsets and the actual impact on employees [30]. Little is known, however, about how engineering experts can incorporate values-driven and ethical approaches into their modelling and development of production systems.

This paper sheds light on the disciplinary axiomatic and epistemic culture of engineering. Engineering is an extremely wide-ranging field of practice, and notions of the human within this may vary widely. The interest is in aspects of technology deployed in the workplace aimed at being implemented in digital (smart) manufacturing processes. Specifically, the focus is on the stage of often incremental innovation that fuels the engineering pipeline with new models or concepts that are discussed within the scientific community.The project started with exploratory interviews [31] in the engineering discipline to understand what informs research activities, what publications are relevant to keep up to date with latest developments and what success in this field looks like. The interviews partially informed key words for a systematic literature review of academic papers. The review focuses on papers within the industrial context of warehousing, where system technologies such as digital twins (DTs), cyber-physical systems (CPSs) and point technologies such as robotics and sensors are considered [13,24,32]. There is ample reflection on warehousing as a context for I4, with publications still being offered in 2023, but less has been done to review this newer area of contribution comprehensively [3,33]. As well as being a test bed for implementing technologies deemed relevant for I5 in manufacturing, engineering and social science research interest has overlapped in the context of warehousing [34,35,36,37]. The papers were assessed through interpretive coding based on intercoder reliability assessments, and focussing on the underlying perception of the role for the human worker in human–technology relations.

## 2. Materials and Methods

### 2.1. Overview of Method

Figure 1 shows the full research process followed along with details of the steps taken under the various stages where appropriate. The sections following on from here unpack these stages in more depth.

### 2.2. Building Interdiscplinarity (Stage 1)

The team came together during a project on AI ethics and the innovation lifecycle involving collaboration with industry, specifically, the National Manufacturing Institute of Scotland (NMIS). The interdisciplinary collaboration on the bid gave some initial, surprising insights into differences in mindsets and workplace cultures that informed the research design in general, and the present review in particular. The research team’s background includes sociology of work, psychology, socio-legal, industrial engineering (smart manufacturing) and computer sciences expertise. The shared interest lies in adding to debates within smart manufacturing and the potential for human-centric approaches developed from within the engineering discipline. Debating AI and ethics, and potential outcomes for humans at work showed an overlap in concerns about the human at work, and shared norms and values in line with I5. However, differences were observed in the ways in which human agency is thought about. While project members from the social sciences would start with job outcomes and impacts on workers, engineering would start with how to improve systems that include human labour, and to avoid harm to workers at the front end of the innovation lifecycle. This observation led to questions around epistemic norms and values within different fields.

The assumption made is that engineering differs in its epistemological approaches from at least some parts of the social sciences. Engineering epistemologies are often framed as based on empiricism, and with a focus on notions of usefulness [38]. In their study on engineering epistemologies, Montfort, Brown and Shinew summarise as follows, “Efforts to distinguish engineering from other disciplines, including the sciences in general, often emphasise two features of the practice of engineering: first, that engineers are more involved with the “real world” than academic scientists; and second, that their interactions with knowledge and certainty are nearly always coloured by the subjective or normative demands of a society”.

This epistemological perspective was taken on board to develop a staged process to see whether prioritisation of usefulness as an orientation towards applied research and related outcomes is conflicting with a values-oriented approach as outlined in I5. A short interview guide was designed to facilitate conversations with engineers on how they engage with the real world, to determine the extent to which they self-orient their interactions toward demands of society, and to identify what they value at work. Given the comprehensive range of human-centric values and practices found in I5, these interviews gave us insights into broader epistemic practices. Interviews were used to understand the sources participants referred to, to keep up to date with developments in their field of expertise, and thus to capture their interactions with knowledge.

### 2.3. Initial Scoping Research (Stage 2)

Six scoping interviews were conducted with the aim of covering the range of the innovation lifecycle in engineering, i.e., the team spoke to doctoral researchers (*n* = 2), a senior engineering academic working within university (*n* = 1), an employee and the CEO of a small business that delivers solutions for 3D enhanced analysis of assemblies (*n* = 2), and an academic working in a centre for manufacturing excellence (*n* = 1). Three topic areas were addressed in the interviews. First, to understand how participants reflect on their job, they were asked to explain their work and responsibilities and to evaluate their job based on things they like and dislike. Second, they were asked what informed their work, and how they kept up to date with latest developments in their field. Last, to understand how participants perceive their own work with regards to broader societal issues and to understand how far the assumption of empiricism as dominant in engineering epistemology is appropriate, the team used a well-established question stemming from empirical research on the social value of jobs asking participants to reflect on whether their job roles “make a meaningful contribution to the world” [39,40].

Participants gave insights into their daily working routines, and the tensions they experience between what they might wish to achieve, and what is possible in the limitations of their resources. Doctoral researchers did not feel as if they needed to compromise on their own values. What was interesting though was no matter what career stage, the access to resources such as cutting-edge technologies made participants compromise. Here, findings from broader literatures on engineering identities are confirmed in that participants are conscious of constraints or even conflicts, but also are “responding to codes of meaning that live at different scales, including contrasting metrics of progress and images of private industry” [41]: p. 393. The notion of “code switching” was visible in the interviews. Boundaries were clearly defined by targets set from industry, or by funders expecting specific outcomes. There was agreement that, if possible, the “users” of innovative processes should be included, though often users were the clients rather than the worker being exposed to the work system. For example, interviewees described how they “model behaviour in” or “create experimental design to simulate human behaviour”.

Interviewees’ reports of what literature or source of knowledge they deem important guided the final selection of literature database for the review. Participants indicated that papers pre-published with arXiv were the most impactful for their daily work. This database is considered to contain the latest state-of-the-art research, in contrast to journal articles which suffer from a time lag from inception to publication. Crucially, arXiv journals were also seen as highly available and so relevant to practitioners as open-access publications. Quality assessment did not seem to be a problem, and it is assumed this is based on the acquired topic expertise in their stage of career. The two doctoral researchers did draw on academic journals in their work, but noted that they used arXiv most regularly. The focus, therefore, is on research available in the arXiv database for this exploratory piece. Subsequent studies could expand on this foundation by considering the wider literature available outside this particular database, but for this paper it is considered that arXiv reflects agreed epistemic culture and embedded publishing practice.

The last question focussing on the perceived relevance for society was positively answered by all participants. Regardless of any tensions which had been previously mentioned in the interview, or even contradictions when it came to resourcing and target setting, unanimously participants agreed they would positively impact on societal progress. Participants framed their relevance around efficiency, reducing extra costs, or minimising poor quality or otherwise undesirable work for human beings. There was wide-ranging agreement on displayed engineering values, mostly combining efficiency, productivity and social progress.

### 2.4. Systematic Literature Review

Systematic reviews are a well-established method in engineering [42] with growing interest in the last 5–10 years [43,44]. This is demonstrated by the breadth and recency of many systematic reviews in the engineering discipline. These reviews covered topics such as additive manufacturing [42], machine learning in a variety of fields [45,46,47,48], studies of gender [49], engineering identity [50,51], lean [52], entrepreneurship [53], and, with particular relevance to this work, human factors [54], sustainable innovation [55,56], engineering education [57,58,59] and ethics interventions [60]. Of course, this is not a meta-review, so the relative prevalence of such research is of mainly contextual relevance. Nevertheless, it does show that there is increasing interest in rigorously understanding various concerns within the discipline, and their basis in literature, including topics related to I5. The approach is based largely around the work of Kitchenham and Charters [61], with supplemental, subject specific considerations provided through the work of Borrego et al. [43,44]. The key steps applied were search term generation, paper search and selection, systematic reduction of the papers selected and structured processes of analysis and assessment. Following their recommendations, the design of this research also avoided overreliance on individual team members, and distributed tasks in a collaborative fashion [44]. Where this work departs from conventional systematic reviews is in the primarily qualitative focus. There is some use of descriptive statistics to support a discussion of the selection process and also to provide some basic insights into the papers marked for inclusion. However, these figures are supplementary to the analysis of sentiments, conceptualisations, evaluations and roles which were observed at play in the papers.

Given the breadth of the engineering field [42,48,56], this study was focussed on a specific topic area that lies at the intersection of what I5 wishes to foster, namely human–machine symbiosis in a context where technology development is focussed on worker safety in high-risk environments with promised potential for future semi-automation [62]. The implementation context of warehousing has been a site for much contemporary exploration of human-centredness for workers. Equally, the warehouse setting has been researched with regards to working conditions, stress and sustainability of workforces. Warehousing is a workplace context where disciplinary research interests overlap, and technologies that are deemed relevant for I5 are implemented. The team is aware of the potential limitation of choosing a non-manufacturing context as, arguably, upskilling might not matter as much in warehousing given the overall business context. With respect to different manufacturing contexts—high volume, low variety vs. low volume, high variety, low value vs. high integrity products—it is assumed that there is no single homogenous activity one can define as manufacturing. It is proposed, however, that warehouse settings align with high variety low value contexts in manufacturing. Equally, warehousing is a key function in many manufacturing systems: the technical problems addressed require the human labourer to be digitally represented, and in this respect, the development process is prone to de-activating workers’ voice.

### 2.5. Search Strategy

This study chooses to focus on one database specifically, the arXiv collection. As noted in Section 2.3, this was chosen due to practitioner relevance. The exploratory interviews suggested that arXiv was used by practitioners as a source of information, whereas papers on academic repositories were more influential with the academics. It is worth noting briefly that arXiv publications have a substantial academic audience, and participants closer to academic environments mentioned they checked these too regularly to gain insights into latest developments in a fast-changing world. ArXiv offers an easily accessible insight into the cutting edge of developments in the field, and it is assumed that many papers are picked up and translated into industry practice. These findings are in line with [63] who showed that engineers when searching for documentary sources relate to time-saving mechanisms.

Search terms were derived from the literature that described the most relevant tools for I4/I5, the research aim and the exploratory interviews. The filtering on title, and later abstract, eliminated papers that were clearly solely technical, with no human–machine related conceptualisation or problem solving. Some papers in the final sample were found to have no human inclusion later, once the full text was read, due to the ambiguity of the term “picker” and whether this role is performed by a human or an automated system, or refers to a means of transport. The first set of terms used was “human”, “employee”, “person” and “staff”. Second, to focus on engineering systems reflecting the human–machine relation from an engineering perspective, a second set of terms was used: “Digital Twin”, “human system” and “human cyber physical system (HCPS)”. Third, the study focussed the analysis on a specific implementation context and searched for papers focussed on “warehouse” or “warehousing”. Multiple searches were conducted employing different focal terms, all with the overall aim of connecting the “human” (human, employee, person, operator 4.0/5.0) and human-oriented technology (HCPS, digital twin, human system). The general form of this search term is expressed below:

(“human*” OR “employee” OR “person” OR “people” OR “staff” OR “human system” OR “cyber physical) AND (“warehouse”)

It is recognised that a wider search of literature databases outside of arXiv may yield useful results under this term however, so this is a worthy consideration for future studies in this area.

### 2.6. Inclusion Criteria

Studies that refer to “the human” (e.g., human labour, employees, resources, beings or factors) were considered for inclusion. In addition, the paper must integrate this consideration alongside a focus on implementation or design of AI, ML, I4, I5 and other related concepts in a warehouse context, as defined in the literature review. To exemplify what is meant here, three distinct papers were discussed. The paper “A Conceptual Reference Model for Human as a Service Provider in Cyber Physical Systems” [64] represents a strong focus on humans: the human is directly included in the focal topic and the main aim of the paper is understanding then integrating the role of the human. This can be contrasted with a paper like “A Proposed Method Using GPU Based SDO To Optimize Retail Warehouses” [65] which stands at the other end of this spectrum, discussing the human primarily in a situating sense in the role of “customer” and “picker”, or as a small component within the broader warehouse system. In between these two extremes, a paper such as “Analysis of Safe Ultrawideband Human-Robot Communication in Automated Collaborative Warehouse” [66] appears, where technology is the focus, but the aim of the paper is to facilitate or accommodate human activity. Only those published in English between 2011 and 2022 were included to reflect the period of emergence of I4 up to the current day developments in I5.

### 2.7. Quality Assessment

Quality criteria were applied to each paper drawing from the criteria used by Dybå, Dingsøyr and Hanssen [67]. The focus of this quality assessment is on the human elements in the research. Most relevant, then, for research purposes were whether the work described the sample or study context, gave insight into data collection methods or analysis (reflecting rigour) and whether relationships between participants and researcher were considered (reflecting credibility). Studies are considered suitable where the details are offered at any level of transparency, rather than whether they were explained to a good degree of detail or not, as this detail is precisely one of the focal points of the analysis. The papers were also assessed for relevance with respect to the aim of this paper; that is, whether the papers align with the practical focus or whether they are theoretical and focussed on internal development of the engineering sciences discipline. The team finally considered whether the papers were aimed at instrumental outcomes or for more general exploratory purposes.

### 2.8. Data Extraction and Synthesis

A data extraction form (Table A3, Appendix C) was created to structure the initial analysis of papers and ensure a systematic reading of each according to the following pre-established criteria: year of publication, key technologies deployed or developed in the paper, degree of human inclusion and the quality assessment categories discussed above.

The scale of human inclusion used is summarised in Table 1. In keeping with the collaborative approach, coding based on this scale was calibrated in a pilot process between three of the research team members. The scale enables quantification of a fundamentally qualitative question, the extent to which papers include or recognise the human element in the context, model or empirical study.

### 2.9. Interpretive Coding 

To understand how human centredness is reflected in epistemic practice, the papers that were extracted and filtered were coded. Papers which were scored at 4 or higher on the scale of human inclusion were analysed as follows. The team segmented the papers in line with pre-defined focal points or categories. The choice was based on the assumption that this set of papers would give us an insight into the most extensive models that took into account the human worker in their experiments and conceptualisation. Engineering sciences is based on models central to knowledge creation. Boon and Knuuttila suggest that models in engineering sciences aim for “scientific understanding of the behaviour of different devices or the properties of diverse materials” [68]. These are modelled based on their functioning in terms of “physical phenomena that produce the proper or improper functioning of the device”. Interpretive coding, it is argued, allows us to gain insights into the quality of inclusion of the human in new technological environments.

This approach, termed interpretive coding, enables us to dissect the language of these papers from a few different perspectives, adopting different lenses to highlight specific aspects of interest [69,70]. The aim here is to consider role attribution, evaluations and underpinning justifications. In other words, we take into account the textual representation of the human and system through the roles they play and attributes attached to that said role, the values, beliefs and attitudes adopted towards humans and systems, as well as the explicitly presented rationale for their inclusion or exclusion described in a particular context [69,71,72]. The aim here is to understand these different systems of values and to determine then compare the sentiments in these discussions.

A hybrid deductive/inductive approach to coding was followed. Initial development of codes was drawn from the sensitising literature and interviews (deduction) [73]. The initial codebook (Table A1, Appendix B) provided a starting point for a process of subsequent inductive development during which attributes were expanded and excluded or dropped through team coding. This approach was taken to ensure that aspects of human centredness and other key theoretical concerns were captured, while also allowing for the generative capacity of iteration. In line with the guidance offered by Kitchenham and Charters, and Borrego, Foster and Froyd, coding was performed by several researchers to ensure reliability and coverage, while the data extraction was conducted by two members of the team and checked by a collaborator [44,61]. The coding step was performed using the software NVivo, a specially designed package intended for analysis of qualitative data [74] and this qualitative content analysis also was conducted by the team [43,61,69]. The final list of codes employed is presented in the appendix (Table A1, Appendix B). Also captured in the descriptive coding process was a collection of in vivo codes (i.e., direct quotes from the papers) which were used to illustrate specific terms, concepts or images employed in the texts analysed.

Three focal points provided initial codes focussing on human-centric aspects:Framing for the problem to be solved in this paper: in this section, coding was initiated against rationales and justifications driving the applied research outcome;Attributes, indicating the roles associated with either technologies or humans and allowing for assessment of the quality of the interaction and collaboration;Values, which reflect evaluations, beliefs and attitudes around humans, machines and the relationship between the two.

Both roles attributed to humans and to technology were intentionally coded. This can be seen reflected in, for example, evaluations of fragility. This code relates to both systems that are fragile and prone to disruption, but also to human beings being seen as physically “fragile” in a highly automated context. Literature presents human fragility as a problem that can be solved with technology, i.e., with more specialised or sentient machines.

## 3. Results

Table 2 summarises the outputs of the selection process, showing the reduction from initial sample to the final 34 selected papers. The initial selection process involved a search and subsequent title-based selection to capture those papers potentially related to the human and machine interface in warehouse contexts. Moving on from the title, works were next screened on the basis of their abstract and excluded if they focussed exclusively on technical systems. This was sometimes unclear due to overlapping terminology. For example, the term “picker” may refer to an automated system or a human operative. Over 40% of the papers (39 out of 92 total) selected from abstracts focussed entirely on systems upon checking of full text and were excluded. Papers also were excluded if they reflected contexts which were not warehousing. Papers were then filtered again using the inclusion/exclusion criteria discussed above. Interpretive coding was applied to the subset of papers particularly relevant to human–machine interaction in the warehouse environment, i.e., with a score of four or greater on the scale of human inclusion (see Table A3 and Table A4, Appendix C for further information on which papers were selected).

### 3.1. Data Extraction Findings

The review identified a recent turn towards the warehouse context in terms of publications on the arXiv database. Despite search parameters extending back to 2011, no relevant results were returned from this range; few were in evidence from 2018 and 2019. As shown in Figure 2, as many relevant papers were published in 2020 as in the preceding two years combined.

When looking at these publications over time, there are a few different perspectives that may be adopted in dissecting the data based around core focal points from the data extraction form. The notion of “publication aim” refers to the purpose of a paper, whether the work is aimed at general exploration and development of a field or instrumental, efficiency-oriented refinement of a technique. In the case where both aspects are applicable, the pieces are seen as instrumental. In terms of publication aim by year (Figure 2), a growth in both exploratory and instrumental pieces can be seen. In terms of raw numbers, the quantity of instrumental pieces has increased more, but as a proportion of the work produced there has been a substantial growth in both. Many of the instrumental pieces had exploratory aspects also, suggesting a general tendency towards developing and improving novel methods, rather than focussing on gaining efficiency in existing industrial settings.

Figure 3 shows a growing proportion of theoretical papers submitted to the database. The change is not so much that empirical pieces become less common in absolute terms; rather they are a smaller proportion of work over time. Aligning with the above, a gradual but meaningful growth in submissions is observable, but one that has been focussed around theoretical pieces which are developed in virtual experimental settings. This growth is representative of a particular epistemological practice, that of engineering sciences as set against the broader community of engineering. This difference may reflect a distinction between design and development versus subsequent logics of implementation and application.

Figure 4 and Figure 5 show the proportion of selected papers which include humans to varying extents. Figure 4 points to a slightly greater prevalence of work which marginalises or only partly includes the human. However, Figure 5 shows that the degree of holistic human inclusion is improving. Indeed, all of the papers with a human inclusion score of four or more have been published within the last three years. This suggests that the discourse of human centrism and I5 more generally is gradually appearing at the level of engineering sciences or model design and development.

Table 3 shows the various technologies employed in or developed through the papers. These were categorised broadly around cyber-physical systems (CPSs), which blend hardware, software and embodied agents in an integrated system; modelling, which related to efforts to construct digital twins; and other digital equivalents to physical systems and management algorithms, which includes those efforts to use software solutions to influence conduct in physical systems. Sensors as a technology were also included, though never as a sole focus in the sample analysed. Rather, these were included in facilitation of modelling and cyber-physical systems.

The review shows a broad spread of technologies, with no category predominating. Few papers in the sample focussed exclusively on one technology, with CPS, management algorithms and modelling contributing around 15% of the total papers each. When these were paired without the inclusion of CPS, primarily theoretical or exploratory pieces which look to model the warehouse context and offer improved algorithms for managing movement of goods, automated systems and, occasionally, people are appearing. Papers which included CPS showed broader consideration of the human. Drawing on the analysis of focal technologies, Table 4 shows the number of papers contributed at each level of human inclusion for each technology (see also Table 2 and associated discussion of this scale). Papers which extend consideration to CPS tend to show a greater degree of human inclusion. This is unsurprising, as the more holistic nature of CPS presents more possibility to include the human as a body, rather than as a variable in modelling. Building on this point, all of the papers at level 2 or below included potentially “abstracted” technologies, namely modelling and management algorithms. These technologies are termed “abstracted” due to the necessary attachment of the physical and digital in notions of CPS and sensors.

### 3.2. Interpretive Coding Results

The interpretive coding was implemented with a first stage of collective reading and establishment of codes. Intercoder reliability was measured and when satisfied, papers were coded by different team members. The results of the coding process are summarised in terms of the code categories, but some examples of relevant quotes are included to show what was coded under specific labels. These quotes are intended to be indicative, rather than summative; that is to say, they represent the kind of messages coded under a specific banner, but the chosen quote merely illustrates part of a set of concepts which were identified under these labels. Each example is tagged with an abbreviated form of the paper title to indicate source i.e., (WVR:2021) is “Warevr: Virtual reality interface for supervision of autonomous robotic system aimed at warehouse stocktaking.” [75]. Interpretive coding of the 11 papers scoring 4 or 5 on the constructed scale of human inclusion shows that, in these research settings, the human is addressed as a generalised human being, reduced to a variable for understanding human intentions which may add value to the technology that is to be integrated. Attributes of the individual worker have to be excluded for modelling reasons. For example, some papers mentioned that sensors would not be used for monitoring heart rates in the training sample to avoid overcomplexity in the model. Typically, experimental modelling approaches-based assumptions on pre-existing datasets are used to justify what is to be excluded.

Arguably, this type of reductionism occurs in most empirical reasoning, including within social sciences. In this context, workers’ agency is reduced to measurable activity, with active agency in the design of projects absent. Instead, if participants are mentioned, the sampling is characteristic of laboratory settings, i.e., with students or researchers mimicking the human warehouse worker in a gamified environment to understand intentions. Human–machine relations, therefore, are considered only at an experimental stage, with the human worker inserted with simulated intentions. To take one example, an experiment was conducted based on a digital twin method to improve human–computer interactions in the warehouse environment with the use of augmented reality. “Participants were students with a background in mechanical engineering, computer science, and robotics. The average participant age was 24.4 (SD = 2.4), with a range of 21–31. The final sample of participants included both novice users and experienced users at drone piloting.” (WVR:2021)

The coding also reflected three initial focal points: 1. framing, 2. attributes, and 3. values (evaluation). Figure 6 provides an overview of the categories which were expanded, along with the codes found to be relevant to the papers.

Framing: The interpretive coding focussing on the framing, or, rationale, highlighted the relevance for efficiency gains in engineering projects. Either papers addressed the costs in general or they claimed to help reduce these by improving the speed and accuracy of the throughflow of commodities. Often, technology is seen as reducing dangerous tasks, hence helping to decrease costs due to accidents at work. Papers generally focussed on the reduction in new tech-induced risks rather than any inherent risks for humans induced by the technology (e.g., work intensification, lack of ergonomic support).Example code:“They use movable racks that can be lifted by small, autonomous robots. By bringing the product to the worker, productivity is increased by a factor of two or more, while simultaneously improving accountability and flexibility.” (HRI2018)Attributes: During the intersubjective coding process, a distinction between human and technology-supported attributes was established. The role for the human is framed around either “collaborator” or as a “service”. None of the coding related to human attributes represented the human as having a voice in relation to decision making, although they were addressed as workers. The notion of the human as operator was absent in this subset of warehouse-focussed papers. In one paper, a smiling face emoji is used to capture the worker in the simulation. The paper does not acknowledge evidence about poor job quality in real world warehouse environments. Instead, the worker seems to be happy, and ends up in a simulation represented as a 1980s computer game character (HIE2019). The attributes, or role for technology is that of an assistant, or, in most cases, of a caretaker. Throughout all papers, the technology was framed as a 24/7 working robot without any need for maintenance. Example code: “It has to be ensured that the worker is assisted and not impeded during work.” (HIE2018)Values (evaluation): A core code emerged in terms of the potential for either the technology or the human as an asset to the process. Technology clearly dominates in this respect, as it was seen as an asset to the process, to the human, and to the firm. The human is mentioned as an asset less frequently, and simply in relation to maintenance work for technology. A second set of values—fallibility, vulnerability and obstacle—appeared far more often when describing a human worker. Consistent with this, technology was framed as supportive in fixing errors occurring in the system and stemming from human action (control and surveillance), while the human was framed by exposing their irrational intentions and unpredictability.

In sum, the results show a role for the human worker modelled to deliver data about intentions and movements, and as necessary to enhance the effectiveness of the technology. Technology has a role as the seamlessly working and reliable tool that needs to be trained to make use of the efficiency gains to function flawlessly around the unreliable human.

## 4. Discussion

The aim of this paper was to consider the extent to which the human is integrated into discussions of advanced operational technologies, whether as an end user or as contributor to the process. The systematic review and interpretive coding of papers applied in a warehousing context confirm the integration of the human through models, introducing human intention as key for efficient and failure-free systems, with sensors being a vital tool for data gathering. One can see a new rationale emerging, with human workers considered not solely as a cost, but now seen as crucial for new technologies to be fully productive. In contrast to earlier iterations of automation, the key debate in smart manufacturing is not about task replacement and upskilling, but about human behaviour that needs to be captured, mostly in the form of human intentions to avoid errors in the system.

Despite this positive direction, the frequently mentioned Operator 5.0 [76,77,78] only exists in very specific settings. There was little evidence supporting I5′s reconciliation of “the human and machine (…to) work in perfect symbiosis” as appraised by policy stakeholders. This vision would mean extracting data about human intention to increase the reliability of the now shared workspace between humans and robots. The idea of collaboration, or collaborative robots working “together” with humans, anthropomorphises workplace relations, with humans in this context guided by data flows, just as robots are, and mitigated by sensors. If symbiosis is to be understood as mutually beneficial interaction between two distinct entities, the benefit to the human workers seems less clear so far in the papers reviewed here. This research has demonstrated the value attributed to the human and the technology in the selected papers. The human worker tends to be evaluated, as fallible, vulnerable, and irrational, therefore, models developed to simulate intentions aim to streamline such irrationality. The technology is framed as the tirelessly working robot that needs feeding with data about the environment to run its programmed tasks to full capacity.

## 5. Conclusions

The findings indicate that a new direction towards the broad aims of Industry 5 (I5) has been taken, with increased effort to include the human in experiments and projects in smart manufacturing, research and development. Engineering has well-established collaborations with management/operations science focussed on risk assessment and safety evaluations and includes broad topic knowledge on human intention modelling to improve the capacity of new technological affordances. I5, however, suggests considering human centredness in a more holistic way, drawing from social sciences more widely to include topics such as collaboration and co-design, and a broad vision of learning factories for skills development. These topics are still to be addressed. The paper demonstrated a stark contrast between the core value of being human-centric as expressed in the I5 vision and the integration of the human in engineering literature. There remains a genuine and pressing challenge to centre the human in a meaningful way within pre-figured epistemic cultures in engineering.

## Figures and Tables

**Figure 1 sensors-23-06416-f001:**
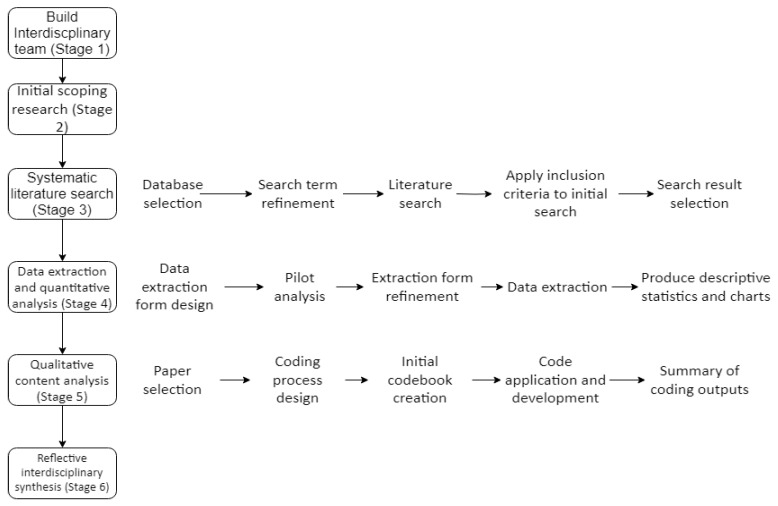
Research Method Process Diagram.

**Figure 2 sensors-23-06416-f002:**
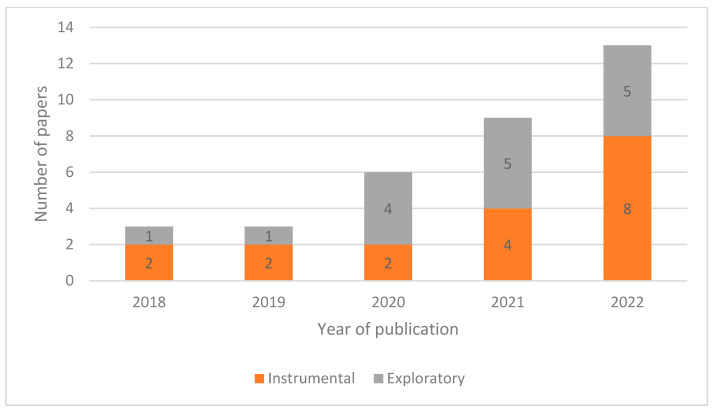
Publication Aim by Year.

**Figure 3 sensors-23-06416-f003:**
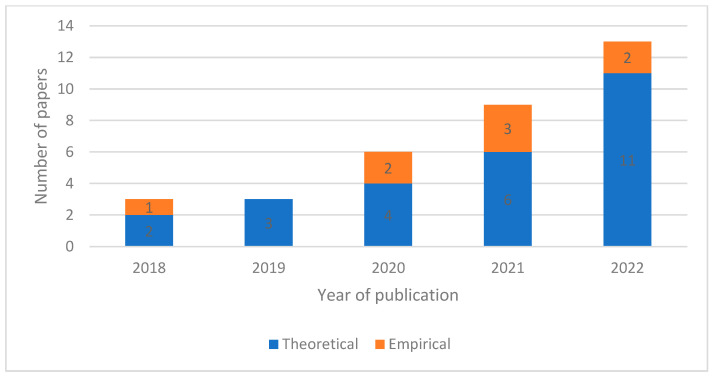
Publication Context by Year.

**Figure 4 sensors-23-06416-f004:**
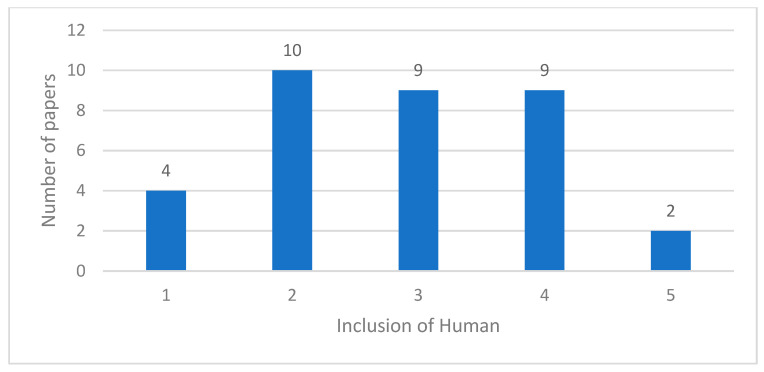
Relevance of Papers.

**Figure 5 sensors-23-06416-f005:**
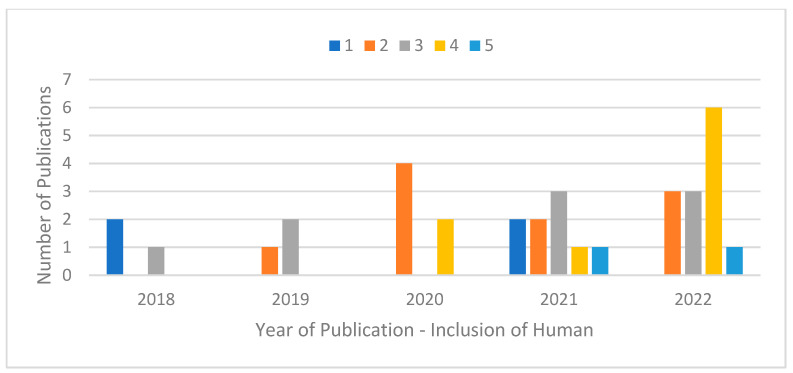
Paper Relevance Over Time.

**Figure 6 sensors-23-06416-f006:**
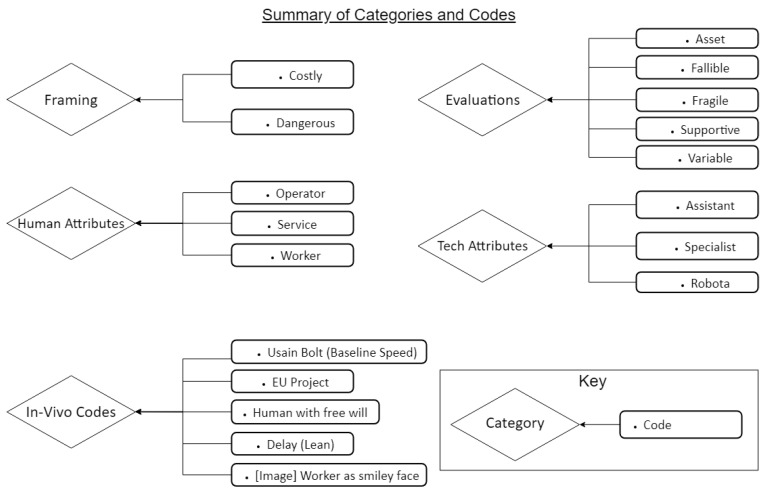
Summary of Categories and Codes.

**Table 1 sensors-23-06416-t001:** Scale of Human Inclusion.

Score	Scope of Inclusion
1	No inclusion
2	Human included in initial framing/as minor variable
3	Human included throughout/as full component in consideration
4	Human included as co-focus of paper or system design
5	Human included as primary focus of paper or system design

**Table 2 sensors-23-06416-t002:** Outcomes of Search Process.

Stage	No.	%
Initial search (articles retrieved though arXiv)	1130	100
Screening of title (excluded if not around human/warehouse)	211	18.7
Screening of abstract (excluded if focussed on technical system only)	94	8.3
Articles eligible after duplicates removed	92	8.1
Articles included in systematic study	34	3
Articles included in the final “coding” analysis	11	1

**Table 3 sensors-23-06416-t003:** Key Technologies Discussed.

Key Technologies Discussed	No.
Modelling, Management Algorithms, CPS	7
Modelling, Management Algorithms	6
Management Algorithms	6
Cyber-physical System (CPS)	5
Modelling	5
Modelling, Sensors	3
CPS, Sensors	1
Management Algorithms, CPS	1

**Table 4 sensors-23-06416-t004:** Human Inclusion in Key Technologies.

	Scale of Human Inclusion (Paper Count)				
Key Technology	1	2	3	4	5
CPS			1		
CPS, Sensors			1		
Cyber-physical System (CPS)			1	2	1
Management Algorithms	1	3		2	
Management Algorithms, CPS			1		
Modelling	1	1		2	1
Modelling, Management Algorithms	2	3	1		
Modelling, Management Algorithms, CPS		2	2	3	
Modelling, Sensors		1	2		

## Data Availability

Publicly available datasets in the form of the arXiv “Computer Science” pre-print paper repository were analysed in this study. This data can be found here: https://info.arxiv.org/help/cs/index.html (accessed 13 April 2023); https://arxiv.org/archive/cs (accessed 12 August 2022).

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
