# Peer review of "Industry 5 and the Human in Human-Centric Manufacturing"

_sensors, 2023, doi:10.3390/s23146416_

Round 1
Reviewer 1 Report
First of all, I would like to read in the manuscript why all the people of the industry or researchers insist on calling it I5.0? This is simply a continuation of I4.0, this is still related to cooperation as I4.0 - please, suggest a proper commentary. Every industrial revolution takes time, and it cannot be slightly several years later… (2011-> I4.0; 2017 - -> I5.0… )
Please, develop the introduction. Especially, since knowledge of I5.0 has not been well established yet.
The methodology graph/description should be included. By the way, although I understand the resigns, references given in ArXiv are not enough for a review manuscript.
Please, be aware that a warehouse term can also be hidden under „logistics centre” or in general as „logistics facility” - please check the works of Prof. Kostrzewski, Dr. FijaÅ‚kowski, etc.
As the Authors mentioned, the term “picker” may refer to an automated system or a human operative, yet it can additionally refer to means of transport such as a lift-truck.
The Authors mentioned: „This Special Issue invited papers to establish new knowledge through technical and sociotechnical contributions that help academia and industry to establish a more holistic approach to digital technologies.” Can you specify which, pls?
Figure 6 should be enlarged - Times New Roman is not a good option for figures’ font. Please, choose a font without sheriffs.
I would like to read more about the process of obtaining results and in general in relation to the quoted references; so far this is a weak point of the manuscript.
Appendix C is hardly possible to read (Tabele C1).
Minor editing of the English language is required, the Authors are native speakers.
Author Response
Please ignore attachment as there was no way to delete the draft
Responses to reviewer 1
Summary revisions
First, we wanted to thank the reviewers for their thoughtful, critical, and supportive feedback and comments. We are convinced that the feedback allowed us to review our paper rigorously, and to revise our contribution to improve the amended technical quality, better articulate the impact, contribution and visualization, and to elaborate on our research process with more clarity and transparency.
In our major revisions, we engaged with all comments. You will find the detailed responses to each reviewer in the section below. The revised resubmission included track changes but we took the liberty to accept format changes with no further implications for the content.
1 The introduction is fully redrafted to address issues concerning
- Meaning of I4/I5
- Research aims and objectives
- Research process
- Methodology
- Novelty/contribution
2 The methodology forming part 2 is outlined to map the stages of the actual research process and now follows a more rigorous structure visualised in a flow diagram (new figure 1). Other major changes include
- Outline of the novelty of our method and a sharpening how it differs from systematic literature reviews.
- Introduces stage 1 ‘Building interdisciplinarity’ as part of the research process and not as an abstract concept in the introduction
- Presents stage 2 interviews in more detail delivering core findings on socio-cultural values
- Engages with paper selection in more detail to avoid misconception, manage expectations, and outline limitations
3 Results section is amended to gain better visualisation results. The coding section explains the choices made, and refers to issues around intercoder reliability as a measure for quality assessment and replication.
4 & 5 Discussion and conclusion focus more on the contribution to debates in the scientific community.
Responses in detail, reviews labelled in line with review submission system.
Detailed responses to reviewer 1
|
Comment |
Authors’ Response |
|
First of all, I would like to read in the manuscript why all the people of the industry or researchers insist on calling it I5.0? This is simply a continuation of I4.0, this is still related to cooperation as I4.0 - please, suggest a proper commentary. Every industrial revolution takes time, and it cannot be slightly several years later… (2011-> I4.0; 2017 - -> I5.0… |
The introduction is completely rewritten and addresses the issues mentioned. We agree with the reviewer that revolutions take time. We clarify that I5 is not claiming to be yet another revolution, but an update. The numbering hence does not go back to the industrial revolutions but is referring more to software update numberings (Windows 10 to Windows 11 etc). |
|
Please, develop the introduction. Especially, since knowledge of I5.0 has not been well established yet.
|
We developed the introduction significantly and established more detail on I5. |
|
The methodology graph/description should be included. By the way, although I understand the resigns, references given in ArXiv are not enough for a review manuscript.
|
We took this comment as an opportunity to visualise our methodology and to clarify on the approach outlining more clearly the stages undertaken that informed the paper.
While we rely on some aspects of a systematic literature review, in this paper, we use the method by applying its tools but are diverting from the classical outcomes. We provide a stronger rationale here, that also helps to support why we use the arXiv papers. Detailing the steps we undertook should support the choices made, our main argument being that arXiv was the repository mentioned by our participants in the scoping stage of the project. We had previously omitted information and findings on this stage and think that its inclusion helps to gain a better insight into the process.
|
|
Please, be aware that a warehouse term can also be hidden under „logistics centre” or in general as „logistics facility” - please check the works of Prof. Kostrzewski, Dr. FijaÅ‚kowski, etc. |
We re-run our search and included the following (333-337): ‘It was noted during the review process that the term “logistics centre” or “logistics facility” would potentially expand the search results obtained. This was trialled and a total of 40 additional papers were uncovered, but the content of all related to outbound logistical challenges located in the wider world as the goods left or were transported between facilities, and so these were outside the scope of the study’s focus on warehousing contexts specifically.’ |
|
As the Authors mentioned, the term “picker” may refer to an automated system or a human operative, yet it can additionally refer to means of transport such as a lift-truck. |
In the filtering process we checked against this issue and selected the human related version. In the revised version of the paper, we clarified this point. |
|
The Authors mentioned: „This Special Issue invited papers to establish new knowledge through technical and sociotechnical contributions that help academia and industry to establish a more holistic approach to digital technologies.” Can you specify which, pls? |
The were not aware the reviewers might not know about the paper being submitted to a special issue, but the comment made us reconsider including this paragraph. We deleted the sentence and assume that with the major changes to the introduction, this point is addressed. |
|
Figure 6 should be enlarged - Times New Roman is not a good option for figures’ font. Please, choose a font without sheriffs. |
Well noted and amended. equally table C.. |
|
I would like to read more about the process of obtaining results and in general in relation to the quoted references; so far this is a weak point of the manuscript. |
We amended the outline for our approach and included a section that addresses our choices. We hope that in clarifying that our contribution is strongly led by assumptions stemming from an epistemic culture perspective that is based on interpretive methods, the section is better situated in the research process. |
Reviewer 2 Report
The main research content presented in the article is a new direction towards the general objectives of Industry 5 with greater effort to include human being in experiments and projects in smart manufacturing, research and development.
The topic is not unique, but it is worthy of researching.
The main proposal is an interdisciplinary systematic literature review and interpretive coding of academic papers.
The deduced conclusions based on the research methods demonstrate an increase in output addressing human activity in modeling and the technologies available to address this concern.
The conclusions are tenable. However, the article is not clear enough what progress has been made compared with the current research results.
The abstract is informative. It reflects the body of the paper.
The introduction provides sufficient background information for readers in the immediate field to understand the problem and the hypotheses.
The text is well arranged and the logic is clear. There are virtually no grammatical errors in this article. The related concepts are introduced clearly. The readability is sufficient.
The article is not clear about the novelty of the approaches or proposals.
The theoretical analysis is sufficient for the purposes of the article.
All figures and tables are clear enough to summarize the results for presentation to the readers. All figures and tables are well referred to in the text.
The reference section is informative. The references are accurate.
Author Response
Please ignore attachment there was no way to change the uploaded draft.
Response to Reviewer 2
Summary revisions:
First, we wanted to thank the reviewers for their thoughtful, critical, and supportive feedback and comments. We are convinced that the feedback allowed us to review our paper rigorously, and to revise our contribution to improve the amended technical quality, better articulate the impact, contribution and visualization, and to elaborate on our research process with more clarity and transparency.
In our major revisions, we engaged with all comments. You will find the detailed responses to each reviewer in the section below. The revised resubmission included track changes but we took the liberty to accept format changes with no further implications for the content.
1 The introduction is fully redrafted to address issues concerning
- Meaning of I4/I5
- Research aims and objectives
- Research process
- Methodology
- Novelty/contribution
2 The methodology forming part 2 is outlined to map the stages of the actual research process and now follows a more rigorous structure visualised in a flow diagram (new figure 1). Other major changes include
- Outline of the novelty of our method and a sharpening how it differs from systematic literature reviews.
- Introduces stage 1 ‘Building interdisciplinarity’ as part of the research process and not as an abstract concept in the introduction
- Presents stage 2 interviews in more detail delivering core findings on socio-cultural values
- Engages with paper selection in more detail to avoid misconception, manage expectations, and outline limitations
3 Results section is amended to gain better visualisation results. The coding section explains the choices made, and refers to issues around intercoder reliability as a measure for quality assessment and replication.
4 & 5 Discussion and conclusion focus more on the contribution to debates in the scientific community.
Responses in detail, reviews labelled in line with review submission system.
Points raised by reviewer 2:
The main research content presented in the article is a new direction towards the general objectives of Industry 5 with greater effort to include human being in experiments and projects in smart manufacturing, research and development. The topic is not unique, but it is worthy of researching. The main proposal is an interdisciplinary systematic literature review and interpretive coding of academic papers. The deduced conclusions based on the research methods demonstrate an increase in output addressing human activity in modeling and the technologies available to address this concern.
The conclusions are tenable. However, the article is not clear enough what progress has been made compared with the current research results.The abstract is informative. It reflects the body of the paper.The introduction provides sufficient background information for readers in the immediate field to understand the problem and the hypotheses. The text is well arranged and the logic is clear. There are virtually no grammatical errors in this article. The related concepts are introduced clearly. The readability is sufficient.The article is not clear about the novelty of the approaches or proposals. The theoretical analysis is sufficient for the purposes of the article.
Comments and actions related to points raised
We wanted to thank the reviewer for the sympathetic summary and appreciation of our work. We sensed though that the core of our paper wasn’t clearly enough communicated, and there was a question about relevance of the paper to inform the wider community of the journal. In our revisions, we focussed on outlining the novelty of our approach.
We hope that in outlining the stages of the project research allow to understand the differences between our approach and a systematic review that includes and interpretive coding stage as addendum. Our aim was to propose an integrated process that uses the tools of a systematic literature review but for a different purpose – to understand epistemic practice and shared values.
Reviewer 3 Report
The authors explore how engineering experts integrate such values into their modelling. 1. To what extent is consideration of the human integrated into discussions of advanced operational technologies? 2. How do these findings inform I5’s quest for human-centric manufacturing reflecting collaboration and co-design between human and machine? Findings demonstrate an increase in output addressing human activity in modelling and the technologies available to address this concern, but the human-centric approach in I5 so far neglects the potential for human agency to increase the effectiveness of manufacturing systems.
The paper is interesting and is well written. However, some major changes are needed:
- - The penultimate paragraph of the introduction section should explain the contributions of the article to the scientific community. After reviewing the document I am unable to clearly identify these contributions. Please check this.
- - The last paragraph of the introduction should explain the content of the following subsections.
- - Fully related with the contribution explanation, the discussion and conclusion section does not show the impact to the scientific community of this work. Please, improve this.
- - The biography contains interesting references. However, references where there is an impact of human-centric artificial intelligence for smart manufacturing applications are lacking.
Castano F., et ál.; Data-Driven Insights on Time-to-Failure of Electromechanical Manufacturing Devices: A Procedure and Case Study (2023) IEEE Transactions on Industrial Informatics, 19 (5), pp. 7190 - 7200, DOI: 10.1109/TII.2022.3216629
Cruz Y.J., et ál., A two-step machine learning approach for dynamic model selection: A case study on a micro milling process (2022) Computers in Industry, 143, DOI: 10.1016/j.compind.2022.103764
In scientific-technical documents, the first-person plural ("we") must not be used. Please use passive or third person voice.
Reviewer 4 Report
The article investigates how engineering professionals incorporate human-focused values into their modeling within the sphere of Industry 5.0 (I5) in digital manufacturing. The authors undertook an interdisciplinary systematic literature review and interpretive coding of scholarly articles to evaluate pertinent I5 technologies. The results highlight an uptick in research output considering human activity in modeling and the technologies that can cater to this area. However, the human-centered approach in I5 currently overlooks human agency's potential to enhance manufacturing systems' productivity. Nevertheless, the overall idea sounds interesting and scientific. The main highlighted points follow such as:
1) While contributing to the discourse on a heavily researched theme, the manuscript in question yields a peripheral impact on the existing body of literature. Furthermore, the presentation and organization of ideas within the document require additional clarification.
This study faces certain limitations, including:
a) The literature review, which confines itself to academic papers, may not encapsulate the comprehensive range of human-centric values and practices found in Industry 5.0.
b) The temporal scope of the literature search, limited to papers published since 2011, with scant results from 2018 and 2019, potentially narrows the breadth of the review.
c) The exploration of the influence of sociocultural factors on incorporating human-centric values in Industry 5.0 needs to be noticed.
c) There needs to be a more in-depth analysis regarding potential challenges and obstacles hindering the application of human-centric values within the context of Industry 5.0.
In summary, while the endeavor to shed light on this pertinent subject is commendable, the findings and contributions seem restricted and demand further refinement.
2) In addition to the points above, it is noteworthy to address that Section 1 of the paper needs to be more verbose, detracting from the clarity of the intended message. The extensive language used in this section may distract readers from the core focus of the study. It would be beneficial to revise this portion for succinctness and better alignment with the main objectives of the article.
3) While comprehensive, The methodology utilized by the authors does not establish a delineated, reproducible routine that third parties could easily follow. This presents a considerable challenge in ascertaining the unique contributions of the manuscript, particularly when juxtaposed with comparable studies. In addition, the logical sequence seems lacking in clarity.
The authors engaged in an interdisciplinary systematic literature review and interpretive coding of scholarly articles to examine pertinent Industry 5.0 (I5) technologies. They employed a scale to measure the degree to which the papers incorporate or acknowledge the human element within their contexts, models, or empirical studies. Furthermore, they coded against the underlying motivations and justifications steering the applied research outcomes and features indicating roles affiliated with either technologies or humans, purportedly to evaluate the quality of the interaction and collaboration. However, the absence of a marked pathway through this process may detract from its efficacy and potential for replication in future work.
4) While the authors applied an interdisciplinary systematic literature review and interpretive coding to examine relevant Industry 5.0 (I5) technologies, the report would benefit significantly from more precise scientometric details, which could have elucidated the proposal more effectively.
In their methodology, they devised a data extraction form to structure the preliminary analysis of the papers, ensuring a systematic examination of each one against pre-set criteria. This form encompassed information like the publication year, core technologies employed or developed in the study, degree of human involvement, and categories for quality assessment. Their scale of human inclusion, summarised in Table 1, facilitated the quantification of an essentially qualitative question regarding the extent to which papers involve or recognize the human element within their context, model, or empirical study. They calibrated the coding based on this scale in a pilot process among three research team members.
However, the method's delineation needs to include in-depth scientometric details that could have provided a clearer understanding of their proposed methodology and its novel contributions. Incorporating such specifications could strengthen the paper's credibility and provide a more concrete foundation for replication and further research.
5) As described, The quality assessment of the papers relies on a somewhat simplistic approach. The authors employed criteria used by Dybå, Dingsøyr, and Hanssen, assessing whether the work detailed the study context or sample, provided insight into data collection methods or analysis (indicative of rigor), and considered the relationships between participants and the researcher (reflective of credibility).
Moreover, the papers' relevance to this study's aim was evaluated — specifically, whether the papers accorded a practical focus or were theoretical, dedicated to the internal development of the engineering sciences discipline. The authors also discerned whether the papers targeted instrumental outcomes or were designed for broader experimental purposes.
While these methods provide some degree of analysis, the approach could be somewhat elementary, failing to exploit more complex or nuanced evaluation techniques thoroughly. A more comprehensive, multi-faceted approach to paper quality and relevance assessment could have added depth to their analysis and contributed to a more robust understanding of the subject.
6) The submission system has multiple versions of the PDF document, creating ambiguity regarding the modifications made from one iteration to the next. Furthermore, the feedback from previous review cycles is absent. For enhanced clarity and traceability of the manuscript's evolution, having a clear record of the changes implemented in each version and a transparent display of prior reviewers' comments and the corresponding responses would be highly beneficial.
7) Furthermore, the graphical representations throughout the manuscript, particularly Figures 6 and 7, do not significantly enhance the work's comprehension. Not only is the resolution of these figures inadequate, making the content difficult to discern, but some images, such as Figure 5, also need to be more precise with excessive information. This overcomplication hinders the viewer's ability to grasp the primary message of the diagrams effectively. Thus, it is recommended that the authors revise these figures to improve both the quality of the images and their relevance to the presented research.
Additionally, it would be beneficial for the authors to thoroughly review the manuscript to eliminate typographical errors and minor grammatical mistakes within the text. These issues, albeit small, may detract from the overall readability and professionalism of the paper. Thus, careful proofreading is recommended to enhance the quality and coherence of the manuscript.
Author Response
Please see attachment Reviewer 4

Round 2
Reviewer 1 Report
As I mentioned before, the references given in ArXiv are not enough for a review manuscript, therefore I strongly suggest the evolution of the Authors' proposal.
Author Response
We understand the concerns raised by the reviewer. In the revised version, we answered the comment in more detail on why and how we make use of the ArXiv database as part of our methodological choice.
The paper is not intended as a systematic literature review of articles which contribute to a scientific body of knowledge. Given that the focus of this paper concerns epistemic practice within an engineering discipline, our method intentionally focused on articles which reflect applied research, that is, the use of knowledge and methods intended to solve practical issues within the chosen context for the study. The ArXiv database was indicated through our exploratory interviews with applied researchers (and our STEM-based co-author) as an appropriate source for such papers. We, therefore, chose the database due to its relevance for practitioners as pre-prints and used the tools of systematic review and qualitative, interpretive coding to understand epistemic practices. We argue that using ArXiv in this way is a legitimate choice and builds a stepping stone for follow up studies. The paper makes a modest claim to gain insights into differences in epistemic practices that is shown using examples of acknowledged publications.
Reviewer 4 Report
I agree the authors proceeded with a significant revision in the paper, and all questions previously mentioned were concerned.
Author Response
Thank you for the feedback and acknowledgement of our effort. We appreciated the thorough and collegiate suggestions that made us reconsider and hence improve the quality of the paper.
Round 3
Reviewer 1 Report
The final decision is in the hands of the Editor.
I understand the Authors' arguments, yet cannot agree with the methodology.
Author Response
We wish to thank the reviewer for their engagement with our argument. Even if we leave in disagreement, we will for sure take their argument with us in our future work on this expert led literature review method. It was a pleasure receiving thoughts from colleagues based in a different discipline, and we see this process as part and parcel of our future interdisicplinary engagement.